# Psychosocial and behavioral correlates with HIV testing among men who have sex with men during the COVID-19 pandemic in China

Lingen Shi[1], Guangxia Liu[2], Gengfeng Fu[1], Nick Zaller[3], Chongyi Wei[4], Cui Yang[5], Hongjing Yan[1] *

1 Jiangsu Provincial Center for Disease Control and Prevention, Nanjing, China, 2 Geriatric Hospital of Nanjing Medical University, Nanjing, China, 3 Fay W. Boozman College of Public Health, University of Arkansas for Medical Sciences, Little Rock, Arkansas, United States of America, 4 Department of Health Behavior, Society, and Policy, Rutgers University School of Public Health, New Brunswick, New Jersey, United States of America, 5 Department of Health, Behavior and Society, Johns Hopkins Bloomberg School of Public Health, Baltimore, Maryland, United States of America

☯ These authors contributed equally to this work.
* yanhongjing@hotmail.com

**Data Availability Statement:** Data from this study are available upon request. Jiangsu Provincial Center for Disease control and Prevention have the whole rights for this data. The data contain

## Abstract

### Objectives

Some of community mitigation efforts on COVID-19 created challenges to ongoing public health programs, including HIV care and prevention services among men who have sex with men (MSM). The goal of the current study was to explore sociodemographic factors and the impact of COVID-19 on HIV testing among Chinese MSM during state-enforced quarantine.

### Methods

We conducted a community based survey between May 1st to June 30th, 2020 on COVID-19 related impacts on HIV testing among 436 China MSM during the COVID-19 state-enforced quarantine.

### Results

One-third (33.7%) of MSM received HIV testing during the quarantine period. Few participants reported difficulty accessing facility-based testing (n = 13, 3.0%) or obtaining HIV self-test kit online (n = 22, 5.0%). However, 12.1% of participants reported being afraid of getting facility-based HIV test due to concerns about the risk of COVID-19. In the multivariate logistic regression model, participants who were married (aOR: 1.89, 95%CI: 1.19–3.01), reported increased quality of sleep (aOR: 2.07, 95%CI: 1.11–3.86), and increased difficulty in accessing health care (aOR: 2.34, 95%CI: 1.37–3.99) were more likely to get an HIV test during the state-enforced quarantine.

### Conclusion

The mitigation measures of COVID-19 have created various barriers to access HIV related prevention services in China, including HIV testing. To mitigate these impacts on HIV

sensitive information. If you want to get the original data for study, please contact with the Corresponding author or ethics committee of Jiangsu Provincial Center for Disease Prevention and Control (Email:439698759@qq.com). The authors had no special access privileges others would not have to the data.

**Funding:** Cui Yang, Nick Zaller and Chongyi Wei's work were funded for this work by National Institute of Health (R21MH118945). CY, NZ and HY conceptualized the study.CY, NZ and CY Wei contributed to data analysis.critically reviewed a revised draft of the manuscript.

**Competing interests:** The authors have declared that no competing interests exist.

prevention and care services, future programs need to address barriers to HIV-related services, such as providing high-quality HIV self-testing. Meanwhile, psychological services or other social services are needed to those experiencing mental distress.

## Introduction

A new coronavirus disease (COVID-19) caused by severe acute respiratory syndrome coronavirus (SARS-Cov-2) was officially reported in Wuhan, China in December 2019 [1]. As of April 12[th], 2021, there were more than 135 million COVID-19 confirmed cases and 2.9 million COVID-19 deaths worldwide [2]. In order to control the pandemic, the Chinese government initiated level one public health emergency response (PHER) nationwide in late January 2020. Based on the existing clinical and epidemiologic evidence on COVID-19, community mitigation efforts were strictly implemented, which included state-enforced quarantine between January to April 2020, social distancing, and a public mask mandate. These measures has effectively flattened the epidemic curve of COVID-19 in China. However, some of these mitigation measures created challenges to ongoing public health programs, including HIV care and prevention services.

Service disruptions to HIV care for people living with HIV/AIDS (PLWHA) were reported in many countries during the COVID-19 pandemic. For instance, UNAIDS conducted an online survey for PLWHA in Wuhan, China and found that 64.2% of PLWHA faced antiretroviral therapy (ART) shortages during the quarantine period [3]. According to a recent rapid assessment of COVID-19 within the HIV health care system in Zimbabwe, more than 50% of PLWHA failed to initiate ART, and 29% were unable to receive viral load tests between April and June 2020 [4]. In order to address the emergent medication supply disruption for PLWHA during COVID-19 quarantine in China, the National Center for AIDS/STD Control and Prevention at the China Center for Disease Control and Prevention (CDC) issued an urgent notice to provide ART through delivering HIV medications via mail services or to transfer HIV cases to the local CDC where PLWHA lived temporarily. However, there was no relevant guidance to address the disruption of HIV prevention services, such as HIV testing. It has been reported that HIV testing dropped off significantly in clinical facilities among key populations worldwide during the COVID-19 pandemic [5, 6].

Men who have sex with men (MSM) continue to be a high risk population for HIV and other sexual transmitted infections [7–9]. Routine HIV testing is an evidence-based HIV prevention approach for MSM at risk of HIV. World Health Organization (WHO) recommends HIV testing at least once every 6 months for individuals at risk of HIV [10]. One of the challenges during the COVID-19 pandemic relates to accessibility of on-site HIV testing and counselling services. In order to address these challenges, local health departments in China encouraged people at risk for HIV to conduct HIV self- testing at home. However, little is known about the barriers and facilitators to accessing HIV testing, either in person or at home, among MSM during the COVID-19 pandemic in China.

The goal of the current study was to explore sociodemographic factors and the impact of COVID-19 on HIV testing among Chinese MSM during state-enforced quarantine. Findings from the current study can inform HIV prevention efforts to better address HIV prevention needs among MSM or other populations at risk of HIV across different settings during the COVID-19 pandemic and in future public health emergencies when HIV-related services are likely to be disrupted.

## Methods

### Study population and recruitment

This survey was conducted concurrently with the annual national HIV sentinel surveillance between April and June 2020 in Jiangsu province, China. The goal of the sentinel surveillance was to monitor the HIV epidemic in key populations in China. Briefly, there were fourteen HIV sentinel sites located in thirteen cities in Jiangsu, and more than 6,000 MSM participated during the 2020 surveillance period. Participants were recruited via multiple methods, including snowball-sampling, online sampling and venue-based sampling [11]. Eligibility of participants included: male; 18 years or older; self-report having oral or anal sex with men within the last 12 months. All eligible participants were invited to one of the survey sites located in the local health department to complete a face-to-face interview and to provide a blood sample for HIV and syphilis testing. We enrolled a subsample of the participants from HIV sentinel surveillance study in three cities Zhenjiang, Wuxi and Suzhou for a supplemental COVID-19 survey.

### Ethics approval and consent to participate

The study was approved by ethics committee of Jiangsu Provincial Center for Disease Prevention and Control (No. of IRB Application: JSJK2019-B012-02). All participants provided written informed consent.

### Measures

The COVID-19 survey included questions about participants' socio-demographics, HIV testing history and barriers to facility-based or self-testing during state-enforced quarantine, i.e., January to April 2020. Participants also reported changes in various aspects of their well-being, psychosocial health, financial difficulties (e.g., paying rent), and sexual health (e.g., number of sexual partners, opportunities of having sex, condom access and use) and substance use (i.e., alcohol and illicit drug use) during the COVID-19 pandemic. Response options included "decreased due to the pandemic," "not changed due to the pandemic," and "increased due to the pandemic" as compared to 6 months before the pandemic.

### Statistics

We reported the frequencies of key measures. We conducted bivariate analysis (Chi-square and unadjusted logistic regression) and forward stepwise multivariate logistic regression, using all independent variables with a p<0.05 in bivariate analyses, to explore the factors related to HIV testing during state-enforced quarantine. All statistical analysis was performed using IBM SPSS STATISTICS (version 19.0, SPSS Inc., Chicago, IL, USA).

## Results

### Study sample and characteristics

A final sample of 436 MSM completed the COVID-19 survey. Table 1 presented the sociodemographic factors, HIV testing during the quarantine period, and various impacts of COVID-19. One-third (33.7%) of MSM received HIV testing during the quarantine period. The majority of participants (97.3%) reported spending at least 40% of their time sheltered at home during the COVID-19 state-enforced quarantine. Few participants reported that they had difficulty accessing facility-based testing (n = 13, 3.0%) or obtaining HIV self-test kit online (n = 22, 5.0%). However, 12.1% of participants reported being afraid of getting facility-based

**Table 1. Locations, barriers for HIV testing during COVID-19 among MSM in Jiangsu Province, China.**

| Characteristics | Number | Proportion |
|---|---|---|
| **Received HIV testing between Jan and Apr (N = 147)** | | |
| **HIV test in** | | |
| CDC | 62 | 42.2 |
| Hospital | 13 | 8.8 |
| CBO | 29 | 19.7 |
| Self-testing | 40 | 27.2 |
| Other | 3 | 2.1 |
| **Barriers for HIV testing during the quarantine period (N = 289)** | | |
| **Trouble accessing facilitate-based HIV testing, such as local CDC or hospital** | | |
| Yes | 13 | 4.5 |
| No | 172 | 59.5 |
| Never tried to get HIV testing | 104 | 36.0 |
| **Trouble accessing HIV self-testing kit online** | | |
| Yes | 22 | 7.6 |
| No | 127 | 43.9 |
| Never try to buying self-testing kit | 140 | 48.5 |
| **Afraid to go to facilitate-based HIV testing (at local CDC or hospital) considering the risk of COVID-19** | | |
| Yes | 35 | 12.1 |
| No | 254 | 87.9 |
| **Time spent during the quarantine period (N = 436)** | | |
| **How much time spent in the house or place lived in between January and April 2020** | | |
| All my time (100%) | 157 | 36.0 |
| Most of my time (70–100%) | 220 | 50.5 |
| Some of my time (40–70%) | 47 | 10.8 |
| Not very much (0–40%) | 12 | 2.7 |

HIV test due to concerns about the risk of COVID-19. Participants reported various impacts on mental health, employment, sexual and substance use behaviors and access to health and social services from the COVID-19 mitigation measures. More than half of MSM (52.5%) reported increased anxiety or stress during COVID-19 quarantine. Nearly half of MSM (45.0%) reported decreased sex partners. Little impacts were observed on access to health and social services such as decreasing opportunities on accessing health care (20.6%) (Table 2).

## Factors associated with having HIV testing during the COVID-19 pandemic

In the unadjusted model (Table 2), participants' age, length of living in the current the city, marital status, various psychosocial, e.g., decreased anxiety or stress, increased quality of sleep, and behavioral impact, e.g., increased opportunities to have sex, increased use of condoms, decreased alcohol consumption from the COVID-19 pandemic were associated with getting an HIV test during the COVID-19 state-enforced quarantine.

In the final multivariate logistic regression model, participants who indicated their relationship status as married were more likely to receive HIV test than those were single (aOR: 1.89, 95%CI: 1.19–3.01) during the state-enforced quarantine. Compared to MSM whose quality of sleep did not change, those who reported increased quality of sleep (aOR: 2.07, 95%CI: 1.11–3.86) were more likely to get an HIV test during the state-enforced quarantine. Compared to

**Table 2. Sociodemographic characteristics and behavioral, economic, and social impact from COVID-19 associated with receiving HIV testing during COVID-19 among MSM in Jiangsu Province, China(N = 436).**

| Characteristics | Total participants (N = 436) | Percent | Getting HIV test (N = 147) | Proportion | P value | Unadjusted OR (95%CI) | Adjusted OR (95%CI) |
|---|---|---|---|---|---|---|---|
| **Demographic characteristics** | | | | | | | |
| **Age** | | | | | | | |
| ≥41 | 95 | 21.8 | 40 | 42.11 | 0.059 | 2.18(1.22–3.90)** | |
| 31–40 | 104 | 23.9 | 38 | 36.54 | | 1.73(0.97–3.07)+ | |
| 25–30 | 117 | 26.8 | 39 | 33.33 | | 1.50(0.85–2.64) | |
| 18–24 | 120 | 27.5 | 30 | 25.00 | | Reference | |
| **Years living in local cities** | | | | | | | |
| >2 years | 300 | 68.8 | 112 | 37.33 | 0.018 | 1.72(1.10–2.70)* | |
| ≤2 year | 136 | 31.2 | 35 | 25.74 | | Reference | |
| **Marital status** | | | | | | | |
| Married | 119 | 27.3 | 52 | 43.70 | 0.007 | 1.81 (1.17–2.80)** | 1.89(1.19–3.01)** |
| Single | 317 | 72.7 | 95 | 29.97 | | Reference | Reference |
| **Impact from COVID-19** | | | | | | | |
| **Anxiety or stress** | | | | | | | |
| Increased | 229 | 52.5 | 76 | 33.19 | 0.016 | 1.11(0.73–1.68) | |
| Decreased | 23 | 5.3 | 14 | 60.87 | | 3.47(1.42–8.47)** | |
| Same | 184 | 42.2 | 57 | 30.98 | | Reference | |
| **Quality of sleep** | | | | | | | |
| Increased | 65 | 14.9 | 37 | 56.92 | < .01 | 3.10(1.78–5.40)*** | 2.07(1.11–3.86)* |
| Decreased | 100 | 22.9 | 29 | 29.00 | | 0.96(0.58–1.59) | 0.86(0.50–1.49) |
| Same | 281 | 62.2 | 81 | 29.89 | | Reference | Reference |
| **Difficulty accessing health care** | | | | | | | |
| Increased | 90 | 20.6 | 45 | 50.00 | < .01 | 2.55(1.58–4.12)*** | 2.34 (1.37–3.99)** |
| Decreased | 30 | 6.9 | 13 | 43.33 | | 1.95(0.91–4.18)+ | 2.08(0.90–4.81)+ |
| Same | 316 | 72.5 | 89 | 28.16 | | Reference | Reference |
| **Number of sex partners** | | | | | | | |
| Increased | 18 | 4.1 | 7 | 38.89 | 0.450 | 1.13(0.42–3.03) | |
| Decreased | 196 | 45.0 | 60 | 30.61 | | 0.78(0.52–1.18) | |
| Same | 222 | 50.9 | 80 | 36.04 | | Reference | |
| **Opportunities to have sex** | | | | | | | |
| Increased | 24 | 5.5 | 15 | 62.50 | 0.004 | 3.08(1.28–7.42)* | 1.75(0.66–4.61) |
| Decreased | 227 | 52.1 | 67 | 29.52 | | 0.77(0.51–1.17) | 0.60(0.38–0.95)* |
| Same | 185 | 42.4 | 65 | 35.14 | | Reference | Reference |
| **Use of dating/hook-up apps or websites to connect virtually with other men** | | | | | | | |
| Increased | 121 | 27.8 | 33 | 27.27 | 0.211 | 0.66(0.40–1.08)+ | |
| Decreased | 114 | 26.2 | 41 | 35.96 | | 0.99(0.61–1.59) | |
| Same | 201 | 46.1 | 73 | 36.32 | | Reference | |
| **Use of dating/hook-up apps or websites to meet other men in person** | | | | | | | |
| Increased | 26 | 6.0 | 7 | 26.92 | 0.378 | 0.62(0.25–1.55) | |
| Decreased | 233 | 53.4 | 74 | 31.76 | | 0.78(0.52–1.18) | |

*(Continued)*

**Table 2.** (Continued)

| Characteristics | Total participants (N = 436) | Percent | Getting HIV test (N = 147) | Proportion | P value | Unadjusted OR (95%CI) | Adjusted OR (95%CI) |
|---|---|---|---|---|---|---|---|
| Same | 177 | 40.6 | 66 | 37.29 | | Reference | |
| **Use of condoms** | | | | | | | |
| Increased | 33 | 7.6 | 17 | 51.52 | 0.038 | 2.35(1.14–4.82)* | |
| Decreased | 53 | 12.1 | 21 | 39.62 | | 1.45(0.80–2.63) | |
| Same | 350 | 80.3 | 109 | 31.14 | | Reference | |
| **Illicit drug use** | | | | | | | |
| Increased | 6 | 1.4 | 3 | 50.00 | 0.593 | 1.95(0.39–9.78) | |
| Decreased | 47 | 10.8 | 14 | 29.79 | | 0.83(0.43–1.60) | |
| Same | 383 | 87.8 | 130 | 33.94 | | Reference | |
| **Alcohol use** | | | | | | | |
| Increased | 66 | 15.2 | 21 | 31.82 | 0.025 | 1.05(0.59–1.87) | |
| Decreased | 79 | 18.1 | 37 | 46.84 | | 1.99(1.20–3.31)** | |
| Same | 291 | 66.7 | 89 | 30.69 | | Reference | |
| **Lost Job** | | | | | | | |
| Yes | 67 | 15.4 | 28 | 41.79 | 0.129 | 1.51(0.89–2.57) | |
| No | 369 | 84.6 | 119 | 32.25 | | Reference | |
| **Lost health insurance** | | | | | | | |
| Yes | 35 | 8.0 | 10 | 6.8 | 0.817 | 0.910(0.409–2.035) | |
| No | 401 | 92.0 | 18 | 6.2 | | Reference | |
| **Lost housing** | | | | | | | |
| Yes | 35 | 8.0 | 15 | 10.2 | 0.233 | 0.654(0.325–1.319) | |
| No | 401 | 92.0 | 20 | 6.9 | | Reference | |

+ p < .10

* p < .05

** p < .01

*** p < .001.

MSM who reported no change in difficulty accessing health care, those who reported increased difficulty in accessing health care (aOR: 2.34, 95%CI: 1.37–3.99) were also more likely to get an HIV test during the state-enforced quarantine. Finally, compared to MSM who reported no change in opportunities to have sex, those who reported decreased opportunities to have sex (aOR 0.60, 95%CI: 0.38–0.95) were less likely to get an HIV test during the state-enforced quarantine.

## Discussion

In this study, we found that just one third of MSM participants (33.7%) received HIV testing during the COVID-19 state-enforced quarantine. According to the annual national HIV sentinel surveillance data, the HIV testing rates among MSM were between 57% to 68% during the same 3-month window from 2016 to 2019 in Jiangsu (unpublished data). A significant decline in HIV testing among MSM during the COVID-19 state-enforced quarantine were observed in our study. The testing rate in our study was much lower than what were reported in studies prior to the COVID-19 pandemic, when the lifetime testing rates were 54% or the testing rate during the past 12 months were 56%-68% [12–14]. Regular HIV testing is a key HIV prevention strategy that can help individuals at risk of HIV to identify their infection earlier in the course of their disease progression, which can in turn lower the risk of transmission if they are

able to initiate ART timely [15, 16]. The Chinese government recommends regular HIV testing every three to six months among high risk populations. With this national guidance, the provincial rate of HIV testing in Zhejiang Province increased from 68.3 to 79.4% during 2013 to 2017 among MSM [13]. The lower prevalence of HIV testing in our study can be due to short-3 month-recall period during state-enforced quarantine. Moreover, similar to findings in other studies on changes in sexual behaviors during the COVID-19 pandemic [17–19], we found participants reporting decrease in opportunities for sex were less likely to get HIV test during the state-enforced quarantine. Lack of opportunities to have sex might have led to lower perceived risk of HIV and needs to have HIV testing.

In China, many MSM tend to get HIV tested at facility-based sites because of these sites provide professional health care services at no cost [20–22]. Although most participants in our study did not report specific barriers to accessing HIV testing during the COVID-19 quarantine period, many of them (12.1%) also expressed concern of being exposed to COVID-19 infection from health facilities, which could have also influenced their decision to get an HIV test at a health facility [23, 24]. Additionally, among those participants who got HIV testing during the COVID-19 quarantine period, only 61.9% tested at facility-based sites. This lower than expected facility-based testing rate may be due to strict quarantine measures implemented in China, as the majority of participants (97.3%) reported spending at least 40% of their time sheltered at home during the COVID-19 state-enforced quarantine. Another study by Odinga and colleagues found that the number of HIV self-tests increased followed by a decline in clinic based HIV testing during COVID-19 quarantine in Kenyan [5]. Though we could not observe any temporal changes of HIV self-testing in this study, availability of HIV self-testing during a pandemic might address some of the barriers associated with strict quarantine measures.

In our multivariate analyses, participants who were married were more likely to get HIV testing during the COVID-19 state-enforced quarantine. Homosexual marriage is not recognized by law in China [25]. MSM experience significant social pressure [26] from their family members' expectations to have a heterosexual marriage [27]. Therefore, Chinese MSM often marry a woman to conceal their homosexuality [28]. Being married might be an indicator of more risky behaviors or taking responsibility of family [29]. In addition, we found participants with increased quality of sleep were more likely to get HIV test. Quality of sleep is associated with mental distress [30]. Mental health problems are common among MSM [7] and it is likely to be execrated during the COVID-19 pandemic due to reduced social connectedness with the LGBTQ community. Mental health problems can adversely affect the uptake of HIV prevention or testing [31–33]. Our results imply that psychological services are much needed among populations at risk during the COVID-19 pandemic. Finally, we found those who reported increased difficulty in accessing health care were also more likely to get an HIV test during the state-enforced quarantine. One of the explanations was participants who got HIV test during the quarantine might be more likely to have observed or experienced service disruptions due to the pandemic.

There were several limitations with this study which should be noted. First, participants of the HIV sentinel surveillance survey were recruited from a convenience sampling (such as snowball sampling, online sampling and venue-based sampling) approach during a global pandemic. Therefore, this sample was not necessarily representative of MSM overall in Jiangsu, China. Second, all data were based on participants' self-report and therefore could be subject to various reporting biases. However, the characteristics of our participants closely mirrored those who participated in the larger HIV sentinel surveillance project (majority less than thirty, local residents, and single) in our study [12, 34, 35]. Studies come from same period with similar populations would be needed to further corroborate our findings.

## Conclusion

We assessed the prevalence of HIV tests among MSM during the COVID-19 state-enforced quarantine in China. The mitigation measures of COVID-19 have created various barriers to access HIV related prevention services in China, including HIV testing. Our study provides timely evidence on the scope of HIV prevention services among Chinese MSM. To mitigate these impacts on HIV prevention and care services, we need to keep or improve HIV related services, such as providing high-quality HIV self-testing. In addition, we should provide psychological services or other social services to those experiencing mental distress.

## Acknowledgments

We thanks for staffs who participate in this study from the local Center Disease Control and Prevention in Wuxi, Suzhou and Zhenjiang.

## Author Contributions

**Conceptualization:** Nick Zaller, Cui Yang, Hongjing Yan.

**Data curation:** Lingen Shi, Guangxia Liu, Cui Yang.

**Formal analysis:** Lingen Shi.

**Funding acquisition:** Cui Yang.

**Writing – original draft:** Lingen Shi.

**Writing – review & editing:** Lingen Shi, Guangxia Liu, Gengfeng Fu, Nick Zaller, Chongyi Wei, Cui Yang, Hongjing Yan.

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
