## [Decision Letter · Decision Letter 0]

17 Nov 2021

PONE-D-21-17838Psychosocial and behavioral correlates with HIV testing among men who have sex with men during the COVID‐19 pandemic in ChinaPLOS ONE

Dear Dr. Ling-en-Shi,

Thank you for submitting your manuscript to PLOS ONE. After careful consideration, we feel that it has merit but does not fully meet PLOS ONE’s publication criteria as it currently stands. Therefore, we invite you to submit a revised version of the manuscript that addresses the points raised during the review process.

We look forward to receiving your revised manuscript.

Kind regards,

Professor Kwasi Torpey, MD PhD MPH

Academic Editor

PLOS ONE

Journal Requirements:

2. Thank you for including your ethics statement: "The study was approved by the local human subjects research institutional review board (No. of IRB Application: JSJK2019-B012-02). All participants provided written informed consent."

4. Please include your tables as part of your main manuscript and remove the individual files. Please note that supplementary tables (should remain/ be uploaded) as separate "supporting information" files"

Reviewers' comments:

Reviewer's Responses to Questions

**Comments to the Author**

1. Is the manuscript technically sound, and do the data support the conclusions?

Reviewer #1: Yes

Reviewer #2: No

2. Has the statistical analysis been performed appropriately and rigorously? 

Reviewer #1: Yes

Reviewer #2: No

3. Have the authors made all data underlying the findings in their manuscript fully available?

Reviewer #1: No

Reviewer #2: No

4. Is the manuscript presented in an intelligible fashion and written in standard English?

Reviewer #1: Yes

Reviewer #2: No

5. Review Comments to the Author

Reviewer #1: The authors evaluated the impact of COVID-19 on HIV testing among MSM in China. Overall, it was a well-written paper and the analyses will sound. However, there are some typos throughout the paper that should be addressed before publication. I have a few other comments as well, noted below.

Methods: Is the survey nationwide or in the Jiangsu province. Line 139 says the province, but line 141 says nationwide.

For eligibility criteria, did the men also have to be HIV negative?

The survey was face-to-face. That could impact response rates during a pandemic. Could you address this in the discussion? What was the response rate? Was there an incentive?

Results: Remember to discuss results in past tense.

Discussion: I think a big limitation in comparing testing rates between your survey and others is the different lengths of time that are assessed. This is addressed briefly, but should be discussed further. When you summarize the results for other studies you should include the time frame (i.e. percent tested within a 12-month period or 6-month period, etc). This is important information when comparing to your 3-month testing rate, since you would expect the proportion to be lower than the proportion testing in past 12-months. You should try to include results from other studies that also ask about a 3-month window for a more comparable comparison.

Tables: In Table 2 you don't need to include both the did test and didn't test columns since you one is the inverse of the other. I would only include the "did test". You can also add a total column and a col percent column, so then most of the info in Table 1 can be shown in Table 2. Table 1 then can just show location of test among those that tested, and the the results of the three questions asked of non-testers. You should also make it clear that those three questions are only asked of the non-testers, so it is clear why it's a smaller sample size.

The description of the location in the title of the table should also be consistent between the two tables.

Reviewer #2: Apart from all other issues in this paper, I do not think it is possible to to analyze the impact of state enforced Covid-19 quarantine on HIV testing if one of the enrollment criteria was "have not taken any HIV test during the past 12 months" (line 145-146). An association between enrollment criteria and the outcome of the study is a serious fallacy. Moreover, the past 12 months also cover the period of covid state enforced quarantine. Hence the impact thereof cannot be evaluated in this study.

6. PLOS authors have the option to publish the peer review history of their article (what does this mean?). If published, this will include your full peer review and any attached files.

Reviewer #1: No

Reviewer #2: No

---

## [Author Response · Author response to Decision Letter 0]

12 Dec 2021

Thank you for the opportunity to revise and resubmit our manuscript, “Psychosocial and behavioral correlates with HIV testing among men who have sex with men during the COVID‐19 pandemic in China”. The recommendations of the reviewer were constructive, and we have made appropriate changes.

---

## [Editor Report · Decision Letter 1]

26 Dec 2021

Psychosocial and behavioral correlates with HIV testing among men who have sex with men during the COVID‐19 pandemic in China

PONE-D-21-17838R1

Dear Dr.Ling-en-Shi,

We’re pleased to inform you that your manuscript has been judged scientifically suitable for publication and will be formally accepted for publication once it meets all outstanding technical requirements.

Kind regards,

Professor Kwasi Torpey, MD PhD MPH

Academic Editor

PLOS ONE
---

## [Editor Report · Acceptance letter]

30 Dec 2021

PONE-D-21-17838R1 

Psychosocial and behavioral correlates with HIV testing among men who have sex with men during the COVID‐19 pandemic in China 

Dear Dr. Shi:

I'm pleased to inform you that your manuscript has been deemed suitable for publication in PLOS ONE. Congratulations! Your manuscript is now with our production department. 

Kind regards, 

on behalf of

Professor Kwasi Torpey 

Academic Editor

PLOS ONE